# Evaluation of Different Methods to Estimate the Transfer of Immunity in Donkey Foals Fed with Colostrum of Good IgG Quality: A Preliminary Study

**DOI:** 10.3390/ani11020507

**Published:** 2021-02-15

**Authors:** Luca Turini, Francesca Bonelli, Irene Nocera, Valentina Meucci, Giuseppe Conte, Micaela Sgorbini

**Affiliations:** 1Department of Veterinary Sciences, University of Pisa, 56124 Pisa, Italy; francesca.bonelli@unipi.it (F.B.); irene.nocera@vet.unipi.it (I.N.); valentina.meucci@unipi.it (V.M.); micaela.sgorbini@unipi.it (M.S.); 2Istituto Zooprofilattico Sperimentale del Lazio e della Toscana ‘M. Aleandri’, 00178 Rome, Italy; 3Centro di Ricerche Agro-ambientali “E. Avanzi”, University of Pisa, 56122 Pisa, Italy; giuseppe.conte@unipi.it

**Keywords:** donkey foals, serum IgG, serum total protein, radial immunodiffusion, refractometer, electrophoresis

## Abstract

**Simple Summary:**

Little is known about the passive transfer of immunity in donkey foals and about the different types of analysis that can be performed to assess it. The aims of the present study were to evaluate the correlation between IgG Serum Radial Immunodiffusion, Electrophoresis Gamma Globulins, Electrophoresis Total Protein and the Serum Total Protein analyzed by Refractometry and by Dry Chemistry Analyzer (Biuret) and to estimate serum IgG concentrations using Serum TP. IgG Serum Radial Immunodiffusion showed a good correlation with Electrophoresis Gamma Globulins and a high correlation with Total Protein Electrophoresis, Biuret and Refractometry. All the tests performed may be a useful to estimate the serum IgG in donkey foals’ blood in the first day of life using a specific equation.

**Abstract:**

The aims of the present study were to evaluate the correlation between IgG Serum Radial Immunodiffusion (SRID), Electrophoresis Gamma Globulins (EGG), Electrophoresis Total Protein (ETP) and the serum total protein (TP) analyzed by refractometry and by a dry chemistry analyzer (Biuret) and to estimate serum IgG concentrations using serum TP. A total of 36 samples collected at four different times (birth, 6, 12, 24 h after birth) from nine Amiata donkey foals were evaluated with SRID, EGG, ETP, serum TP Biuret and refractometry. SRID IgG concentration increased significantly over time until T12. Serum TP analyzed with refractometry, electrophoresis and Biuret showed a statistically significant difference between T0 and T6 vs. T12 and T24. A good or strong correlation was found between different tests performed. Equations to quantify serum IgG were created and can be used for estimating the donkey foals’ serum IgG in the first day of life. Serum TP refractometry showed a high correlation with SRID IgG (0.91) which may be a particularly useful and economic instrument to estimate the transfer of immunity in donkey foals during the first day of life. Further studies evaluating a high number of animals are needed in order to set specific cut-off values.

## 1. Introduction

Despite donkey foals being born with a higher level of circulating antibodies compared to the equine foals, they still required a good amount of high quality colostrum due to the type of placentation [1,2,3,4,5]. Donkey placenta is diffuse and epitheliochorial with numerous microplacentomes consisting of a fetal microcotyledonary and a maternal microcaruncular part [1]. A well-managed passive transfer of immunity allows the donkey foal to achieve an adequate serum IgG concentration at 24h of life. The most common consequences connected with low quality colostrum in neonatal foals is sepsis which can manifest in several different ways, such as bacteremia, pneumonia, enterocolitis, omphalitis/omphalophlebitis, meningoencephalitis and septic arthritis [6]. Despite the relevance of these topics, very few studies evaluated the peripartum field in donkey and no cut-off values have been reported for the quantification of the transfer of immunity in donkey foals. Based on data about equine foals, transfer of immunity is considered to be adequate when serum IgG are ≥ 8 g/l, while partial failure of transfer is considered with serum IgG levels between 4 and 8 g/L and total failure of transfer with < 4 g/L [7,8,9,10]. An early detection of partial or total failure of transfer of immunity is crucial for the health of the donkey foals [7,8,9].

In equine foals, several tests were used to assess the transfer of immunity and the IgG concentrations evaluated by the single radial immunodiffusion (SRID) is considered the gold standard test. However, the SRID test is expensive, needing 18–24 h to obtain results and it is susceptible to human error during the measurement of the ring diameter. Results can also vary between the different commercial kits used [11,12,13]. Electrophoresis can be an appropriate test for the evaluation of the transfer of immunity, but it is rarely used. Serum electrophoresis results need only few hours for being ready and they do not differ between different kits as SRID. Moreover, a recent study showed a good correlation between SRID and electrophoretic gamma globulins [14]. Other tests used which provide quick results are the zinc sulfate turbidity, the glutaraldehyde coagulation, and the ELISA-based tests, such as the Snap Foal IgG test [13,15]. Refractometry represents an inexpensive, rapid, and accurate test for the failure of transfer of passive immunity (FTPI) evaluation [16,17,18,19].

The use of refractometry for serum total protein (TP) evaluation is mainly used in dairy farms for calves’ immunity transfer assessment [16] while it is rarely used in foals as an indicator of transfer of immunity [17,18]. Only a few studies have been done to evaluate the use of the refractometry as indicator of FTPI in foals because the value of serum TP can be overestimated by high levels of glucose, urea and plasma lipids [19]. Novel, more accurate and less influenced methods to quantify serum TP has been found recently; serum IgG could be estimated with the use of serum protein concentrations measured with an automated chemistry analyzer in foals [17]. Compared to serum TP evaluated with a refractometer the automatic analyzer presented the advantage of not being influenced by the concentrations of plasma lipids (cholesterol, lipoproteins), glucose, urea, or excessive ethylenediaminetetracetic acid (EDTA). The disadvantages are the cost and the low feasibility in the field [17]. 

Despite the fact that the literature is wide for equine foals, there is a lack of knowledge about the assessment of transfer of immunity in donkey foals. Studies in different fields of donkey foal’s neonatology [5,20,21] showed that differences between these two species exist and are crucial. These considerations make the investigation about other methods, such as electrophoresis, refractometry and Biuret analysis of some potential interest for the rapid assessment of passive transfer of maternal antibodies in neonatal donkey foals. Thus, the aims of the present study were to assess IgG and serum TP at different sampling times, to evaluate the correlation between IgG SRID, Electrophoresis Gamma Globulins (EGG), Electrophoresis Total Protein (ETP) and the serum TP analyzed by refractometry and by dry chemistry analyzer (Biuret) and to estimate serum IgG concentrations using serum TP.

## 2. Materials and Methods 

### 2.1. Animals

Nine Amiata donkey foals were included in this study. All the donkey foals belong to the Regional studfarm “Le Bandite di Scarlino” (Grosseto, Italy). The Ethical Committee of the University of Pisa (Organismo Preposto Benessere Animale, OPBA) approved the study with protocol number 22/19. An owner’s written consent was also obtained. All the donkey foals were born at the Veterinary Teaching Hospital “Mario Modenato”, Department of Veterinary Sciences, University of Pisa during the foaling season 2019–2020. Immediately after birth all the donkey foals were evaluated with a dedicated APGAR score [22]. All the donkey foals were fed by their own dam with a good quality of colostrum within the first 3 h of life [23,24]. Colostrum has been evaluated as reported in Turini et al. 2020 [3]. The mean ± standard deviation of the colostrum IgG concentration of the 9 Jennies before the foal’s nursing was 89.32 ± 26.43 g/L. A complete physical examination as reported in [25] was performed at each sampling time in order to assess the health status of the donkey foals. Special attention was made to those parameters evaluating the absence of infection.

### 2.2. Sampling Procedures

Ten mL of blood were collected immediately after birth, before the first colostrum feeding, (T0), and at 6 (T6), 12 (T12) and 24 (T24) h after birth from jugular vein of each donkey foal in order to assess IgG and TP concentrations. Samples were harvested in red-top Vacutainer tubes (10-mL BD Vacutainer glass serum tube, silicone-coated; Becton Dickinson and Co., Franklin Lakes, NJ, USA) and immediately centrifuged (Legend RT, Sorvall; ThermoFisher Scientific Inc., Waltham, MA, USA) at 1.565× *g* for 15 min in order to collect the serum. The serum samples were then stored at −20 °C until the evaluation. The day of the analysis, the serum samples were defrosted at room temperature and vortexed for 10 s just before the dilution.

### 2.3. Determination of Serum IgG and Total Protein

Studies have shown a strong homologue between donkey IgG and horse IgG [26,27]. Because of the lack of a specific SRID for the donkey, and because of the close similarities of the immune systems of horses and donkeys, a SRID assay specific for horse was used to analyze IgG colostrum concentrations [26,27]. All the serum samples were analyzed in a single batch. SRID was performed using a Horse IgG IDRing (R) Test (IDBiotech, Issoire, France). Test results were determined by comparison with a standard curve prepared using equine immunoglobulin standards (25, 50, 100 and 200 µg/mL) supplied with the Kit. Serum samples were diluted 1/150 before IgG determination, as suggested by the manufactory instruction, because of their high IgG concentrations.

Serum TP was measured using a temperature-compensating digital refractometer (AR200; Reichert Analytical Instruments, Reichert Inc., Depew, NY, USA). Electrophoresis gamma globulin (EGG) and electrophoresis total protein (ETP) have been analyzed with a fully automated electrophoresis instrument (Pretty Interlab, Sebia Company, Rome, Italy). Concentrations of TP were also measured using a dry chemistry analyser (SAT450, Assel, Aprilia, Italy) by biuret assay according to the manufacturer’s instructions.

### 2.4. Statistic Analysis

A G-power analysis was performed in order to calculate the minimum number of animals that should be included. G-power analysis showed that the minimum number of animals needed was 4, considering an effect size (Cohen coefficient) of 0.8, an alpha value of 0.05 and a power of 0.8. 

Data for normal distribution were evaluated by the Shapiro-Wilk test. Data on the IgG SRID, EGG, TP Biuret, TP Refractometry and ETP were analyzed by the following linear model, using JMP software (SAS Institute Inc., Cary, NC, USA):y_ijz_ = μ + T_i_ + A_z_ + ε_ijz_
where y_ijz_ = dependent variables; Ti= fixed effect of the ith time of sampling (T0, T6, T12, T24); A_z_ = random effect of the zth animal (9 levels); ε_ijz_ = random residual.

Least-square means with their standard errors were reported, and treatment effects were declared significant at *p* < 0.05.

Correlations between IgG SRID, EGG, TP Biuret, TP Refractometry and ETP were evaluated calculating Pearson’s coefficient; only correlations with a p-value of the linear model below 0.05 were considered as significant.

The linear contrasts were tested in the first model by the t-test with Tukey’s adjustment within each parity level.

## 3. Results

A total of 36 samples (four sampling times × nine donkey foals) were analyzed. None of the included donkey foals showed signs of disease or discomfort during the time of the study. The temperature of the donkey foals evaluated of each sampling time was always within the normal range [25]. Data were normally distributed.

The mean ± standard deviation for each method used at different collection time were reported in Table 1.

Correlation analysis results and *p*-value between the different methods used for the evaluation of donkey foals’ serum immunoglobulins and TP were reported in Table 2.

A strong significant relation between serum IgG SRID and serum TP Biuret (*p* < 0.0001, R^2^ = 0.80) has been found. The value of serum IgG can be calculated using a serum TP Biuret with the following formula: SRID IgG (g/dL) = −0.90 + 0.42 × TP Biuret value (g/dL). Additionally, the relation between serum IgG SRID and serum TP Refractometry was highly significant (*p* < 0.0001, R^2^ = 0.82). The value of serum IgG can be calculated using a serum TP Refractometry with the following formula: SRID IgG (g/dL) = −0.85 + 0.43 × TP Refractometry value (g/dL). A strong significant relation between serum IgG SRID and serum ETP (*p* < 0.0001, R^2^ = 0.76) has been found. Serum IgG can be calculated using a serum TP Biuret with the following formula: SRID IgG (g/dL) = −0.80 + 0.4 × ETP value (g/dL). The confidence interval of each methods investigated was reported in Figure 1.

## 4. Discussion

In foals, SRID is considered the gold standard method to quantify serum IgG concentrations [11,28,29]. However, the SRID method could have a high individual error based on different interpretations of the precipitin ring. Additionally, results can be influenced by time of incubation and temperature which might be different based on the SRID assays from varies manufacturers [11,13,29]. Moreover, due to the lack of studies about validation of SRID analysis in donkey foals, this technique could not represent the gold standard one for this species. The aims of the present study were to assess IgG and serum TP at different sampling times, to evaluate the correlations between IgG SRID, EGG, ETP and the serum TP analyzed by refractometry and by dry chemistry analyzer and to estimate serum IgG concentration using serum TP.

SRID IgG concentration increased significantly over time until T12, remaining stable between T12 and T24. Serum TP analyzed with refractometry, electrophoresis and dry chemistry analyzer showed a statistically significant difference between T0 and T6 vs. T12 and T24, while no differences were found between T0 vs. T6 and T12 vs. T24. Studies performed in equine foals recommended determining serum IgG levels at around 18 h of life [30] or even earlier [31]. The small intestine remains permeable to macromolecules, most importantly IgG, during the first 18–24 h after birth allowing the absorption of ingested colostrum antibodies [32]. The most important risk factors for FPTI in the foal include the delay in suckling time, feeding foal with a low quality of colostrum or the lack of absorption of the IgG [32]. Thus, knowing as early as possible the level of passive transfer of immunity in donkey foals leads to a prompt veterinary intervention in case of need and to a better prognosis [32]. From the results of the present study, we can assume that an early assessment of transfer of immunity made by total protein evaluation may be done at around 12 h of life even in donkey foals [30,31].

Gamma globulins evaluated by electrophoresis showed an increasing pattern over time, however no statistically significant differences were found between different sampling times. These results were in line with equine foals in which EGG concentration increased during the first hour of life, especially between 4 and 8 h after birth [31]. However, the lack of a statistically significant increase over time is surprising and seems in contrast with other findings [15]. Despite the number of animals included in the present study being in line with what was requested by the G-power analysis, increasing the population for further studies may be indicated. Additionally, EGG performed in donkeys showed a separation of IgGa from IgGb, while horses presented IgGab and IgGc subtypes [27]. This characteristic related to the species may have influenced our results. 

IgG SRID showed a good correlation with EGG and a high correlation with TP Biuret, TP Refractometry and ETP. Very few studies reported methods comparison for the evaluation of transfer of immunity in equine foals, and the differences between methods and statistical analysis approach used made it difficult to compare results between papers. The coefficient of correlation between IgG SRID and TP Refractometry was slightly higher in donkey foals (0.91) compared to equine foals (0.85 and 0.73, respectively) [11,18]. The correlation between EGG and TP Biuret was lower (0.50) in our donkey foals population compared to a study performed in equine foals (0.93) [14]. This difference may be due to the evaluation of TP Biuret made in the present study instead of the Biuret gamma globulins performed in the research of the Swiss colleagues. This difference in methods may explain the discrepancy between results. The same study also compared IgG SRID to total globulins made by Biuret analysis which found a correlation of 0.79. Our results showed a correlation of 0.89 between IgG SRID and TP Biuret leading to preferring the evaluation of total protein compared to total globulins for immunity transfer assessment in donkey foals [14]. 

Results from the present study showed that all the methods used for the serum TP assessment presented a high correlation between each other and could be used to estimate serum IgG with the formula found. However, due to the high dispersion around the regression curve the prediction of plasma IgG concentration can only be made within a large prediction interval. The predominant immunoglobulin in equids colostrum is IgG, and FTPI is assessed by the evaluation of IgG in foals’ serum during the first hours of life [33,34,35]. As previously said, this is possible only by using laboratory techniques which may not be feasible for the owners under field conditions. The evaluation of serum TP levels is usually considered more practical because is cheap, ready to use and not influenced by operator skills, or environmental conditions (e.g., temperature, humidity, etc.) [19]. Due to these characteristics, TP evaluated by refractometer is a technique used by farmers for the evaluation of ruminant neonates [16]. Different equations have been calculated to estimate serum IgG from total proteins value obtained by different methods. These equations could be very suitable for owners or clinicians especially when there is no possibility to evaluate IgG levels by using the gold standard technique.

## 5. Conclusions

This study suggests a good correlation between IgG SRID, EGG, Biuret TP, Refractometry TP and ETP. Based on our results, serum TP refractometry showed the highest correlation with SRID IgG, along with TP Biuret and ETP. Due to its feasibility in the field, TP refractometry may be a particularly useful and economic instrument to estimate the transfer of immunity in donkey foals’ during the first day of life. Further studies evaluating a higher number of animals would be needed in order to set specific cut-off values.

## Figures and Tables

**Figure 1 animals-11-00507-f001:**
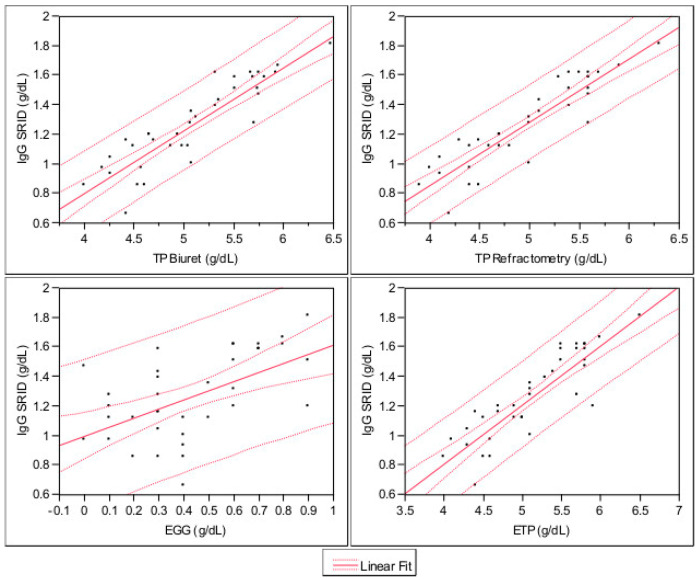
Confidence interval of each methods investigated. Internal dashed lines = Prediction lines. External dashed lines = Confidence lines. Legend: IgG SRID—immunoglobulin type G evaluated by the single radial immunodiffusion; TP Refractometry– total protein evaluated by refractometry; EGG—electrophoresis gamma globulin; ETP—electrophoresis total protein.

**Table 1 animals-11-00507-t001:** Results concerning serum IgG concentrations or total protein concentration evaluated by different methods, expressed in mean and standard deviation, in a population of nine donkey foals (total of 36 samples) at different collection time. Legend: IgG—immunoglobulin type G; SRID—serum radial immunodiffusion; EGG—electrophoresis gamma globulin; TP—total protein; ETP—electrophoresis total protein. Legend: A ≠ B ≠ C *p* < 0.05 between values on the same column; *** = *p*-value < 0.001.

Hours Post-Partum.	*n*	IgG SRID (g/dL)	EGG (g/dL)	TP Biuret (g/dL)	TP Refractometry (g/dL)	ETP (g/dL)
0	9	0.9 ± 0.12 ^C^	0.28 ± 0.15 ^A^	4.44 ± 0.31 ^B^	4.29 ± 0.33 ^B^	4.43 ± 0.32 ^B^
6	9	1.18 ± 0.08 ^B^	0.41 ± 0.25 ^A^	4.80 ± 0.24 ^B^	4.67 ± 0.26 ^B^	4.94 ± 0.44 ^B^
12	9	1.49 ± 0.16 ^A^	0.48 ± 0.31 ^A^	5.58 ± 0.39 ^A^	5.44 ± 0.36 ^A^	5.59 ± 0.38 ^A^
24	9	1.49 ± 0.20 ^A^	0.57 ± 0.25^A^	5.57 ± 0.44 ^A^	5.46 ± 0.42 ^A^	5.59 ± 0.45 ^A^
All time points	36	1.26 ± 0.29	0.43 ± 0.26	5.1 ± 0.6	4.96 ± 0.61	5.14 ± 0.62
SE		0.049	0.082	0.118	0.116	0.134
*p*-value		***	Ns	***	***	***

**Table 2 animals-11-00507-t002:** Values concerning the coefficient of correlation between different methods for the evaluation of immunoglobulins type G and serum total protein in a population of nine donkeys. Legend: IgG—immunoglobulin type G; SRID—serum radial immunodiffusion; EGG—electrophoresis gamma globulin; TP—total protein; ETP—electrophoresis total protein; ** = *p*-value between 0.01 and 0.001; *** = *p*-value < 0.001.

Methods	IgG SRID (g/dL)	EGG (g/dL)	TP Biuret (g/dL)	TP Refractometry (g/dL)	ETP (g/dL)
IgG SRID (g/dL)	1.00	** 0.56	*** 0.89	*** 0.91	*** 0.87
EGG (g/dL)		1.00	** 0.50	** 0.52	** 0.59
TP Biuret (g/dL)			1.00	*** 0.99	*** 0.94
TP Refractometry (g/dL)				1.00	*** 0.95
ETP (g/dL)					1.00

## Data Availability

The data presented in this study are available in article.

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
