# Peer review of "Evaluation of Different Methods to Estimate the Transfer of Immunity in Donkey Foals Fed with Colostrum of Good IgG Quality: A Preliminary Study"

_animals, 2021, doi:10.3390/ani11020507_

Round 1
Reviewer 1 Report
This manuscript has been improved significantly compared to its original version. The authors made substantial changes to the Abstract, Materials and Methods, and to the Discussion. Despite this, this reviewer still has minor comments related to Table 2… please move the (*) above the correlation number (** 0.56).
Author Response
Reviewer 1
Major comments
Despite this, this reviewer still has minor comments related to Table 2… please move the (*) above the correlation number (** 0.56).
AU: done (see table 1).

Reviewer 2 Report
“Evaluation of different methods to estimate the transfer of immunity in donkey foals fed with high
colostrum of high IgG quality: a preliminary study” by Luca Turini et al. revised version
this paper presents an interesting work in which different tools are applied and compared to estimate the transfer of passive immunity in Donkey foals.
I would first like to thank the authors to have answered most of my request on the first version of the manuscript.
I however still have some minor comments or recommendations to improve the paper.
* Abstract: line 36 : change “higher” to "high" and give the estimated correlation coefficient
* Line 137 and 154: G-power: please precise what you mean by “an effect size of 0.8”. An effect size of what? In addition, this, including the result of the G power, should be in the M&M section, in order to justify your sample size.
* Line 156-157: "The mean±standard deviation of the quality of colostrum of the 9 Jennies before the foal’s nursing was 89.32±26.43 g/L": while the mean value corresponds to high IgG concentrations, the sd estimate suggests that all colostrum samples were not of “high” quality. Am I right ? Moreover, please change “of the quality of colostrum” to “of the IgG concentration”. Finally, this might be in the M&M section
* Table 1: column EGG. A superscript A should be added to all figures, as there is no significant difference between sampling time
* Figure 1: 2 pairs of dashed lines are presented in each panel around the linear prediction curve. Please explain what they stand for (confidence / prediction intervals?)
* Figure 1: dispersion around the regression curve is high and therefore the prediction of plasma IgG concentration can only be made within a large prediction interval. This should be discussed.
-- end of review
L’email a bien été copiéAuthor Response
Reviewer 2
Major comments
Abstract: line 36 : change “higher” to "high" and give the estimated correlation coefficient
AU: done (see line 35).
Line 137 and 154: G-power: please precise what you mean by “an effect size of 0.8”. An effect size of what? In addition, this, including the result of the G power, should be in the M&M section, in order to justify your sample size.
AU: The effect size is an estimation of sample variability based on Cohen coefficient calculation. High value of Cohen coefficient is related to high variability that reduce the number of observations needed for analysis (see lines 136-137).
Line 156-157: "The mean±standard deviation of the quality of colostrum of the 9 Jennies before the foal’s nursing was 89.32±26.43 g/L": while the mean value corresponds to high IgG concentrations, the sd estimate suggests that all colostrum samples were not of “high” quality. Am I right ? Moreover, please change “of the quality of colostrum” to “of the IgG concentration”. Finally, this might be in the M&M section.
AU: It was a draft error. It is a good quality of colostrum. The title and the main text have been changed (see lines 4, 99-101)
Table 1: column EGG. A superscript A should be added to all figures, as there is no significant difference between sampling time.
AU: done (see table 1)
Figure 1: 2 pairs of dashed lines are presented in each panel around the linear prediction curve. Please explain what they stand for (confidence / prediction intervals?)
AU: Internal dashed lines = Prediction lines. External dashed lines = Confidence lines. The main text has been modified (see lines 192-195).
Figure 1: dispersion around the regression curve is high and therefore the prediction of plasma IgG concentration can only be made within a large prediction interval. This should be discussed.
AU: The main text has been modified (see lines 243-244).

Reviewer 3 Report
Dear Authors,
I am not sure if you understand my comments correctly. The goal of your study was to evaluate the transfer of immunity in donkeys measured by different methods. Thus, You should measure the Ab concentration using different methods in colostrum and the serum. Then the correlation should be found.
In addition, Authors collected only 9 animals. However, I find that donkeys foals are not so unique. Thus, bigger group should be collected.
According to those comments I encourage Authors to expand the group and prepare more measurements.
Author Response
Reviewer 3
Major comments
I am not sure if you understand my comments correctly. The goal of your study was to evaluate the transfer of immunity in donkeys measured by different methods. Thus, you should measure the Ab concentration using different methods in colostrum and the serum. Then the correlation should be found.
AU: The aim of our study was to assess the correlation between different methods used for IgG and total protein evaluation in serum, not in colostrum.
In addition, Authors collected only 9 animals. However, I find that donkeys foals are not so unique. Thus, bigger group should be collected. According to those comments I encourage Authors to expand the group and prepare more measurements.
AU: We thank the reviewer for the suggestion. The editor will decide.

This manuscript is a resubmission of an earlier submission. The following is a list of the peer review reports and author responses from that submission.
Round 1
Reviewer 1 Report
The manuscript evaluated 5 methods for measuring IgG/TP in the serum of donkey foals. Several studies already evaluated these assays in horse foals. However, there is some novelty related to the use of these methods in donkey foals. The manuscript is reasonably well explained and structured. However, the English standard needs to be improved to make the report easier to read and remove unnecessary repeated information. Therefore, I will delay my decision until I check the revised version.
Major comments
- Authors collected samples from only 9 donkey foals; therefore, details about sample size calculation should be provided to ensure that this number of samples is enough to give power for the statistical analysis performed.
- The authors mentioned that the data were normally distributed, how they assessed the normality of only 9 samples.
- The manuscript's objective was to assess the correlation between IgG measured by RID and STP measured by other methods. However, no objectives were reported about the assessment of IgG and STP at different sampling times.
- Several published papers evaluated each method's diagnostic performance in horse foals and reached the same conclusion, so authors should justify and add details about the novelty of their work like why there is a need to assess these methods in donkey foals?
- There is no much information in the manuscript, so that it should be considered as a short communication.
Minor comments
Line 34: delete the from “showed the higher correlation” and add “showed a higher correlation.”
Line 63: correct “failure of passive transfer” to “failure of transfer of passive immunity (FTPI)” …. FTPI is the correct abbreviation that is currently used to describe the low IgG concentration in foal serum.
Line 102: what is the standard IgG range that used to develop the curve?
Line 103: How samples were diluted, and why the dilution is so high (1/150)? Did the samples lie outside the standard IgG range, retested?
Line 124: authors used Pearson’s correlation, so how the normality of the data evaluated?
Line 124: P-value should be 0.05 not 0.005?
Author Response
Reviewer 1
Major comments
Authors collected samples from only 9 donkey foals; therefore, details about sample size calculation should be provided to ensure that this number of samples is enough to give power for the statistical analysis performed.
AU: We calculated the sample size using the G-power analysis. The result of the analysis showed that the minimum number of animals is 4. We did the statistical test for repeated measures number used an effect size of 0.8 (based on our experience), an alpha value of 0.05 and a power of 0.8. We added the information in the main text (see lines 137-138, 154-155).
The authors mentioned that the data were normally distributed, how they assessed the normality of only 9 samples.
AU: We did a Shapiro test for the 36 samples of each method. Shapiro test showed that data were normally distributed. We added the information in the main text (see line 138).
The manuscript's objective was to assess the correlation between IgG measured by RID and STP measured by other methods. However, no objectives were reported about the assessment of IgG and STP at different sampling times.
AU: The main text has been modified (see lines 88-89).
Several published papers evaluated each method's diagnostic performance in horse foals and reached the same conclusion, so authors should justify and add details about the novelty of their work like why there is a need to assess these methods in donkey foals?
AU: Done (see lines 84-88).
There is no much information in the manuscript, so that it should be considered as a short communication.
AU: We thank the reviewer for the suggestion. The editor will decide.
Minor comments
Line 34: delete the from “showed the higher correlation” and add “showed a higher correlation.”
AU: Done (see line 35).
Line 63: correct “failure of passive transfer” to “failure of transfer of passive immunity (FTPI)” …. FTPI is the correct abbreviation that is currently used to describe the low IgG concentration in foal serum.
AU: Done (see line 70).
Line 102: what is the standard IgG range that used to develop the curve?
AU: Done (see lines 124-126).
Line 103: How samples were diluted, and why the dilution is so high (1/150)? Did the samples lie outside the standard IgG range, retested?
AU: Samples were diluted with PBS as suggested for serum samples in the kit instructions. No, the samples did not lie outside the standard IgG range (see lines 125-126).
Line 124: authors used Pearson’s correlation, so how the normality of the data evaluated?
AU: We did a Shapiro test for the 36 samples of each method. Shapiro test showed that data were normally distributed. We added the information in the main text (see line 138).
Line 124: P-value should be 0.05 not 0.005?
AU: It was a draft error. The main text has been modified (see line 149).

Reviewer 2 Report
L’email a bien été copié “Evaluation of different methods to estimate the transfer of immunity in donkey foals fed with highquality of colostrum IgG: a preliminary study” by Luca Turini et al.
this paper presents an interesting work in which different tools are applied and compared to estimate the transfer of passive immunity in Donkey foals.
My main comment is that this study is conducted on a very small sample (n=9), which prevents any generalization. Although the title indicates that this is a preliminary study, I am not sure that it is worth publishing the results obtained on such a small sample. The results presented here could be part of a larger study in which decision thresholds regarding the failure of transfer of passive immunity are provided, including the diagnostic accuracy of different methods.
Would the manuscript, in spite of all, be considered for publication, I still have some comments that may help the authors improve the paper and fulfill high research standards.
Major comments
Title: should it be “…donkey foals fed with colostrum of high IgG quality”? Not sure that it is worth mentioning that the colostrum has a high IgG concentration.
Table 2 is essentially the same as table 1 and all information could be gathered in one table.
Lines 158 – 165 and 207-210: not sure that the regression equations are worth been mentioned and detailed. Moreover, you cannot stand “that the value of serum IgG can be calculated” without mentioning any confidence interval, which is missing. I however guess that this confidence interval might be large, because of the small sample size. Please, therefore, delete these regression equations and the corresponding discussion section or add a confidence interval. This could also be plotted on a graph, with all methods together.
Lines 184-86: how can you explain that EGG did not show any statistical increase over time?
Lines 194-195: this is strange that your EGG estimates correlate so badly with all other methods? Was there something wrong with your electrophoresis method? If not, how do you explain the better results obtained in equine foals by Tscheschlok et al (2017)?
Lines 196-199: I do not understand these two sentences as they do not make sense or are difficult to understand.
Line 214-215: you can not stand that “total protein refractometry showed the higher correlation with SRID IgG”. That correlation between TP biuret and SRID IgG is 0.89 with is similar to 0.91, and the correlation between TP biuret and TP refractometry is 0.99, which means that both methods give almost the same results.
Minor comments
"biureto" should be changed to biuret wherever it appears
Section 2.3: please mention here that EGG = electrophoresis gamma globulin; ETP = electrophoresis total protein, and so on, so that the reader does not need to find it in tables or at the end of the introduction.
Line 124: should it be "...p-value of the linear model below 0.05"? Not 0.005
Table 3: only half of the table is needed because the information above the diagonal is the same as the information below.
Line 215: “showed the highest” instead “higher”; a “and” is missing after SRID IgG.
Author Response
Reviewer 2
Major comments
Title: should it be “…donkey foals fed with colostrum of high IgG quality”? Not sure that it is worth mentioning that the colostrum has a high IgG concentration.
AU: Done (see lines 2-5).
Table 2 is essentially the same as table 1 and all information could be gathered in one table.
AU: Done (see lines 164-180)
Lines 158 – 165 and 207-210: not sure that the regression equations are worth been mentioned and detailed. Moreover, you cannot stand “that the value of serum IgG can be calculated” without mentioning any confidence interval, which is missing. I however guess that this confidence interval might be large, because of the small sample size. Please, therefore, delete these regression equations and the corresponding discussion section or add a confidence interval. This could also be plotted on a graph, with all methods together.
AU: A figure regarding the confidence interval has been added to the main text (see lines 198-203).
Lines 184-86: how can you explain that EGG did not show any statistical increase over time?
AU: EGG showed an increase during the different collection time. The lack of statistical significance over time may be due to the differences in IgG patterns for the EGG analysis in donkey foals and for the number of samples evaluated in our study. The discussion has been improved (see lines 236-241).
Lines 194-195: this is strange that your EGG estimates correlate so badly with all other methods? Was there something wrong with your electrophoresis method? If not, how do you explain the better results obtained in equine foals by Tscheschlok et al (2017)?
AU: See the answer to the previous comment and the improved discussion (see lines 236-241).
Lines 196-199: I do not understand these two sentences as they do not make sense or are difficult to understand.
AU: The main text has been modified (see lines 249-250).
Line 214-215: you can not stand that “total protein refractometry showed the higher correlation with SRID IgG”. That correlation between TP biuret and SRID IgG is 0.89 with is similar to 0.91, and the correlation between TP biuret and TP refractometry is 0.99, which means that both methods give almost the same results.
AU: The conclusion has been modified (see lines 276-277).
Minor comments
"biureto" should be changed to biuret wherever it appears
AU: Done.
Section 2.3: please mention here that EGG = electrophoresis gamma globulin; ETP = electrophoresis total protein, and so on, so that the reader does not need to find it in tables or at the end of the introduction.
AU: Done (see lines 129-130).
Line 124: should it be "...p-value of the linear model below 0.05"? Not 0.005
AU: It was a draft error. The main text has been modified (see line 149).
Table 3: only half of the table is needed because the information above the diagonal is the same as the information below.
AU: Table 3 has been modified (see lines 183-189).
Line 215: “showed the highest” instead “higher”; a “and” is missing after SRID IgG.
AU: Done (see line 276).

Reviewer 3 Report
This is an interesting study with new information which can be very useful in veterinary and breading practice. It can be accepted to be published in Animals after revisions. However, it includes some weaknesses than have to be discussed properly. Also the Results should be more reader friendly because are hard to follow up.
Introduction
Although interesting, the rationale or motivation for this research is not clear enough. In the introduction, the authors should be more specific and more explanation should be added.
L43: more detailed information about placentation in donkeys should be added.
L45: consequences connected with low quality colostrum ex. which infections are the most common in donkeys foals with insufficient passive transfer should be added.
L53: the citation should be added
L63: in which species? Calfs?
L64: the citation should be added about refractometry in foals
L65: "increases” should be changed to “high” level of GLU, UREA and LIPIDS?
L67: the citation should be added
L69: what are the advantages and disadvantages of using automated analyzers in that case?
Materials and Methods
How did you managed the blood samples before analyzing
L85: the measurements of colostrum quality should be performed and the results should be added
L85: what do You mean by complete physical examination? Exanimated parameters should be emphasised. The rectal temperature should be added leading to exclusion of infection influence on the results.
L89: the time 0 consider the time before colostrum consumption? It should be emphasized.
L99: the citation should be added
L124: 0.005 or 0.05?
Results
Some results are difficult to understand, not clearly presented.
L136: change “and” to “or”
Table 2 is not reader friendly. It is not clear for which parameters the p value is dedicated. Mabey using the figures will be more suitable.
Table 3 is also hard to read.
L161: was “highly” significant
Discussion
The discussion should be more extensive. There is lack of information which are important for the reader. The discussion which method is mostly recommended should be added with explanation why.
L170: describe time and temperature influence
L171-L176: it should be transfer to introduction
L181: the more detailed explanation of process leading to acquire protective level of Ig should be added. Also the processes which may influence on fail in passive transport.
L184: the advantages of fast Ig evaluation should be added.
Conclusions
The interpretation of the results is insufficient, and in certain aspects, bibliographic references are lacking to support the authors' assertions. As a consequence, the conclusions are not supported by the results obtained. This part of the manuscript needs further elaboration and conviction in the interpretations / statements / conclusions presented.
Author Response
Reviewer 3
Introduction
Although interesting, the rationale or motivation for this research is not clear enough. In the introduction, the authors should be more specific and more explanation should be added.
AU: The introduction has been revised (see the introduction section).
L43: more detailed information about placentation in donkeys should be added.
AU: Information about placentation in donkeys have been added to the main text (see lines 45-46).
L45: consequences connected with low quality colostrum ex. which infections are the most common in donkeys foals with insufficient passive transfer should be added.
AU: The main text has been modified (see lines 48-50).
L53: the citation should be added
AU: Done (see lines 57, 62).
L63: in which species? Calfs?
AU: The main text has been modified (see lines 71-73).
L64: the citation should be added about refractometry in foals
AU: Done (see line 76)
L65: "increases” should be changed to “high” level of GLU, UREA and LIPIDS?
AU: Done (see line 75)
L67: the citation should be added
AU: The text has been modified (see lines 82)
L69: what are the advantages and disadvantages of using automated analyzers in that case?
AU: The text has been modified (see lines 79-82).
Materials and Methods
How did you managed the blood samples before analyzing
AU: The main text has been modified (see lines 115-116).
L85: the measurements of colostrum quality should be performed and the results should be added
AU: Done (see lines 103-104).
L85: what do You mean by complete physical examination? Exanimated parameters should be emphasised. The rectal temperature should be added leading to exclusion of infection influence on the results.
AU: The main text has been modified (see lines 104-106, 158-160).
L89: the time 0 consider the time before colostrum consumption? It should be emphasized.
AU: Done (see line 109).
L99: the citation should be added
AU: Done (see line 122)
L124: 0.005 or 0.05?
AU: It was a draft error. The main text has been modified (see line 149).
Results
Some results are difficult to understand, not clearly presented.
AU: The results section has been changed.
L136: change “and” to “or”
AU: Done (see line 165).
Table 2 is not reader friendly. It is not clear for which parameters the p value is dedicated. Mabey using the figures will be more suitable.
AU: Table 2 has been modified to make the table clearer (see lines 164-170).
Table 3 is also hard to read.
AU: Table 3 has been modified to make the table clearer (see lines 181-189).
L161: was “highly” significant
AU: Done (see line 194).
Discussion
The discussion should be more extensive. There is lack of information which are important for the reader. The discussion which method is mostly recommended should be added with explanation why.
AU: Discussion has been improved (see discussion section).
L170: describe time and temperature influence
AU: Done (see lines 207-209).
L171-L176: it should be transfer to introduction
AU: The main text has been modified (see lines 210-213).
L181: the more detailed explanation of process leading to acquire protective level of Ig should be added. Also the processes which may influence on fail in passive transport.
AU: The main text has been modified (see lines 221-225).
L184: the advantages of fast Ig evaluation should be added.
AU: The main text has been modified (see lines 225-229).
Conclusions
The interpretation of the results is insufficient, and in certain aspects, bibliographic references are lacking to support the authors' assertions. As a consequence, the conclusions are not supported by the results obtained. This part of the manuscript needs further elaboration and conviction in the interpretations / statements / conclusions presented.
AU: Done (see lines 275-280).
